# Anxiety among Central American Migrants in Mexico: A Cumulative Vulnerability

**DOI:** 10.3390/ijerph20064899

**Published:** 2023-03-10

**Authors:** Shoshana Berenzon Gorn, Nayelhi Saavedra, Ietza Bojorquez, Geoffrey Reed, Milton L. Wainberg, María Elena Medina-Mora

**Affiliations:** 1Dirección de Investigaciones Epidemiológicas y Psicosociales, Instituto Nacional de Psiquiatría Ramon de la Fuente Muñíz, Mexico City 14370, Mexico; 2Departamento de Estudios de Población, El Colegio de la Frontera Norte, Tijuana 22560, Mexico; 3Department of Psychiatry, Columbia University, New York, NY 10032, USA; 4Centro de Investigación en Salud Mental Global INPRFM, UNAM, Mexico City 14370, Mexico; 5Facultad de Psicología, UNAM, Mexico City 04510, Mexico; 6Seminario de Estudios Sobre la Globalidad, Facultad de Medicina, UNAM, Mexico City 04510, Mexico

**Keywords:** migrants in transit, Central Americans, emotional distress, accumulated vulnerability, Mexico

## Abstract

Migration exposes Central American migrants, particularly those who migrate without documents, to a range of incidents, dangers, and risks that increase their vulnerability to anxiety symptoms. In most cases, the poverty, conflict, and violence they experience in their countries of origin are compounded by the unpredictable conditions of their journey through Mexico. The objective of this study was to explore the association between the presence of emotional discomfort and the experience of various vulnerabilities from the perspective of a group of Central American migrants in transit through Mexico. This is a descriptive, mixed-methods study (QUALI-QUAN). During the qualitative phase, thirty-five migrants were interviewed (twenty in Mexico City and six in Tijuana). During the quantitative phase, a questionnaire was administered to 217 migrants in shelters in Tijuana. An analysis of the subjects’ accounts yielded various factors associated with stress and anxiety, which were divided into five main groups: (1) precarious conditions during the journey through Mexico, (2) rejection and abuse due to their identity, (3) abuse by Mexican authorities, (4) violence by criminal organizations, and (5) waiting time before being able to continue their journey. The interaction of various vulnerabilities predisposes individuals to present emotional discomfort, such as anxiety. Migrants who reported experiencing three or more vulnerabilities presented the highest percentages of anxiety symptoms.

## 1. Introduction 

In the 1980s, emigration from countries such as Honduras, Guatemala, El Salvador, and Nicaragua ceased to be intraregional, focusing instead on the United States and Canada. People no longer left their countries of origin for purely economic reasons but were forced to migrate. These changes were caused by the dynamics of inequality and social breakdown, resulting from socioeconomic deterioration and the increase in structural violence committed by criminal groups, exacerbated by the indifference of the region’s authorities. One of the immediate repercussions of these situations on the population has been the increase in emigration [1,2]. The majority of Central Americans who emigrate do so under multiple adverse conditions, embarking on their journeys with limited funds, weak social networks, and inaccurate knowledge of the obstacles they may encounter on the migratory routes through Mexico [3].

These adverse conditions can be exacerbated by criminal activities targeting migrants or the extreme weather conditions in northern Mexico. All of the above are potential risks that occur during the journey of those entering the south of the country seeking to reach the United States border. In a study of migrants, González and Aikin [4] identified the risks along the way, grouping them into three sets: natural or weather-related, public insecurity involving criminal acts, and institutional ones associated with migration controls. The authors concluded that the risks combined with factors such as gender, age and migratory experience give rise to various degrees of vulnerability, whether extreme, severe, or moderate. Since vulnerability is the human condition of being hurt [5], in this article, we analyze the relationship between the vulnerabilities to which migrants are subject and the damage to their emotional and mental health they may cause.

To this end, we take up the proposal of “accumulated vulnerabilities,” through which the convergence of two or more adverse circumstances is analyzed at certain points in time [6]. The analysis of accumulated vulnerabilities is a proposal from the field of public health that describes the way various economic, environmental, social, cultural and personal adversities reinforce each other and predispose individuals and communities to health problems [7,8].

Difficulties encountered along the way include the stiffening of securitization and containment policies in Mexico and the United States. The first consequence of these policies has been an increase in the number of arrests and deportations of people lacking the necessary documentation, exposing them to abuses and violations of the rights of migrants by Mexican authorities [9,10,11,12,13]. 

The next consequence of containment policies is that to avoid immigration controls, people on the move are forced to travel along more dangerous routes since, in the first instance, the supply of basic resources for survival, such as water, food and shelter, not to mention medical aid [14,15], is non-existent or unaffordable. Doctors Without Borders [3] note that over 40% of Central American migrants requiring medical care at some point during their journey through Mexico are unable to access health services. 

At the same time, it has been documented that on these alternative routes, migrants are more exposed to the activities of criminal groups [13,16]. A report by Doctors Without Borders [3], for which 480 Central American migrants seeking care at this organization’s care centers were interviewed, showed that during their passage through Mexico, 57.3% were exposed to some type of violence, 39.2% suffered a violent incident, and 27.3% were threatened or extorted. 

Another consequence of border outsourcing policies, compounded by the implementation of Title 42 due to the COVID-19 pandemic in 2020, is that migrants’ stay in Mexico is longer than they had anticipated. This permanence has been called “forced immobility” or entrapment [17,18]. During this stage, people find it necessary to look for work, but their lack of documentation means that they are usually unable to secure employment or end up engaging in activities that involve labor exploitation [8,10].

### Consequences of Vulnerabilities

These situations affect the mental health of people on the move in a variety of ways and to differing degrees. Symptoms ranging from worry, emotional discomfort, and anxiety to post-traumatic stress disorder have been reported. Temores-Alcántara et al. [19] reported that migrants felt “worried”, “scared”, and “nervous” because of the lack of basic resources such as food and accommodation. 

Willen et al. [20] pointed out that depression, anxiety disorders, and chronic pain were common in the transit stage and, in many cases, were related to exposure to violence or other traumatic events, such as fear of being detained or returned to their country of origin. Having been a victim of violence alone accounted for 78% of the mental health consultations of migrants on their way through Mexico, 42% of whom presented symptoms of anxiety, 35% depression, and 10% post-traumatic symptoms [3]. It is important to note that these results are not representative of all Central American migrants in transit through Mexico since they were drawn from those who sought care from Doctors Without Borders. 

A prolonged or longer-than-expected stay in the country of transit increases the probability of presenting emotional discomfort. Requena et al. [21] found that Central American migrants focus their energies on the “American dream”, which helps them cope with all the adversities they suffer while in transit through Mexico. However, the longer they are stranded in this country, the less they are able to cope with these adversities and the greater their risk of developing mental disorders becomes. 

During the time migrants are forced to remain in a town or city, they seek accommodation in shelters or temporary settlements and rely on food either from these shelters or charities [17,18]. These situations can create the sensation of having lost control over their lives and weakens their self-esteem [22]. In addition, as has been studied among other types of vulnerable populations, becoming “spectators of their needs and mere consumers of the care afforded them” probably damages social honor and dignity [23].

The objective of this study was to explore the relationship between the presence of emotional discomfort and the experience of various vulnerabilities from the perspective of a group of Central American migrants in transit through Mexico. 

Emotional discomfort is understood as the perception of moods a person recognizes and expresses, such as worry, sadness, anger, fear, stress, and anxiety. 

## 2. Materials and Methods 

This is a descriptive exploratory study in which a mixed methods model (QUALI → QUAN) was used [24]. Semi-structured interviews were conducted with thirty-five Central American migrants living in shelters in Mexico City and Tijuana, Baja California. In addition, a survey was conducted of 220 migrants in fourteen shelters in Tijuana. Table 1 summarizes the data collection process and presents the general characteristics of the subjects.

### 2.1. Data Analysis

Interviews were analyzed by establishing categories and subcategories using Kvale’s [26] (1994) meaning categorization technique, which made it possible to extract smaller discursive units that were easier to compare. Two people coded the interviews separately and identified emerging themes that were grouped into categories that had not previously been considered. The encodings of each analyst were subsequently compared, and discrepancies in the categorization were discussed and resolved. The categorized discursive units were subsequently contrasted with the complete texts of each interview. 

For the data obtained from the survey, descriptive analyses were performed. The statistical analysis was conducted using the SPSS 21 and STATA 13 programs. Because the sample size of LGBTI migrants was small and did not allow for statistical analysis, this group was not included in the quantitative results. However, some of their narrations were subsequently revisited.

For the description of vulnerabilities, the following categories were established: 

*1. Precarious conditions during the journey through Mexico:* Refers to the lack or insufficiency of money and, therefore, food, water, and other supplies during their journey. This was compounded by limited access to basic services such as accommodation, toilets, showers, and medical care, as well as the lack of immigration documents and information on transit routes, human rights, and administrative processes.

*2. Rejection and abuse because of their identity:* Includes factors related to gender-sex identity, socioeconomic class, ethnonational characteristics, and undocumented status.

*3. Abuse by Mexican Authorities:* Refers to economic extortion, mistreatment and non-compliance with the legal framework, or violations of the provisions enshrined in migration laws by immigration authorities, federal and state police, and the army.

*4. Violence by criminal organizations:* Acts performed by these organizations with varying degrees of violence (such as assaults, beatings, sexual attacks, kidnapping, murder, and coercion). 

*5. Waiting time before being able to continue their journey:* The period when migrants must remain in a town or city due to factors curtailing their desire to move. 

### 2.2. Ethical Care

The research was approved (CEI/C/061/2020) by the Research Ethics Committee of the Ramón de la Fuente National Institute of Psychiatry (IRB00006105). It was considered a minimal-risk project, so informed consent was obtained verbally. In both phases, authorization was requested from each of the subjects, who were informed of the objectives of the project, the data collection techniques, and the confidential nature of the information. Authorization to release the information was also requested.

## 3. Results

Results were organized according to the five vulnerabilities identified, and in each one, extracts from the subjects’ remarks are presented, together with graphs showing the survey data. 

### 3.1. Precarious Conditions during the Journey through Mexico

Most subjects remarked that they had left their place of origin in a hurry, without being able to raise enough money or prepare for the trip by stocking up on the necessary supplies. Nor did they have sufficient information about the unpredictability of the trip. When they entered Mexico, they were forced to walk for hours in the hot sun without proper footwear. They reduced their food and water consumption, and some remarked that they had been forced to spend the night outdoors without warm clothes or blankets. After spending many days under these conditions, they said they felt significant physical wear and tear. Mario, a Honduran, summed this up as follows, “The road is very rough, it’s hard on your feet, and there is a lot of tiredness and dehydration”.

According to the subjects, the lack of money is a constant because not having immigration documents makes it extremely difficult for them to secure employment. For example, regarding accommodation, the majority said that they were forced to alternate periods living in shelters with periods living on the streets. 

These precarious conditions were also observed in the responses to the survey on food insecurity (food insecurity is defined as a situation in which people lack access to adequate, safe, nutritious food to meet the daily nutritional requirements for a healthy life (Sánchez et al., 2019)). In this respect, 35% of migrants reported that they had not eaten for a whole day during their last stay in Tijuana, while 58.8% declared that they had stopped eating breakfast, lunch or dinner. These data reflect a severe level of food insecurity. Moreover, one in six migrants stated that they had eaten less than they should have, placing them in the category of medium or moderate insecurity (see Table 2).

Subjects associated food deprivation with negative moods or emotions such as “depression” and constant “worry”. Apropos of this, Samuel (from El Salvador) remarked: “You get depressed because you don’t have the money to buy a plate of food, because you can’t afford to buy a plate of food. You are worried because there are times when there is none. And there is also a problem of work, which means you can’t buy what you need, personal items for yourself.”

For those traveling with small children, the lack of food was an even more pressing problem. Edgar, a Honduran, shared the following: “Yesterday I went out to cry in the corner over there, outside the shelter, because at home, for as long as I can remember, there has always been a fruit basket in the middle of the table in my house… but now we are here, yesterday I did not have a penny in my pocket, I was trying to get milk for my children…With five children it is not easy, you have to take care of them.” 

The lack of money that caused housing problems and food deprivation created a state of permanent vulnerability for the subjects, which was compounded by another condition with a negative impact: their identity characteristics. 

### 3.2. Rejection and Abuse Because of Their Identity

Gender-sex identity, socioeconomic class, nationality, ethnicity, and undocumented status are characteristics that caused migrants to experience stigma, rejection, and even abuse and violence. These attitudes were displayed by Mexicans, other migrants, and even the staff of institutions related to migration issues. 

Regarding nationality, some subjects reported that in various places they had passed through, those with eateries, bathrooms or showers, or motorcycle taxi drivers charged them more than they did Mexicans. 

Omar, a Honduran, said that “in the place we were renting, because we were migrants, it was quite expensive. We had some neighbors who were Mexican, and they paid about $1200 pesos, but we were charged $2500.”

Regarding the rejection due to gender identity, those who experienced it repeatedly and most aggressively were transgender women, who recounted incidences of discrimination and mistreatment. Ana (a Honduran) mentioned the following: “I think that in Mexico I have had a lot of discrimination. I felt bad because… I came supposedly to look for a new life in Mexico to feel good. They had told me that there was no discrimination here, but I have suffered a lot of discrimination, I went to work but they didn’t gave me a job because I was trans.”

Verbal attacks and mistreatment due to their skin color and ethnic characteristics were experienced by men and women of any age, including children. Jordán (a Honduran) remarked, “Many Mexicans are good, but some are racist. They treat you badly, and they call you a ‘bloody raccoon from Honduras or trash’ and that makes you feel bad.” 

Esperanza (a Honduran), who is the mother of two children, said that she felt “depressed” because “... I have two children who are black and they are the only blacks and they treat them very badly. They say things to them and we hear them but we don’t say anything because after all, I am not in my country.” 

When the interviewees narrated these experiences and were asked how they had felt, the most common answer was “bad.” When asked to be more specific, they said they felt “sad”, “angry” or “awful”. 

In the survey, being rejected due to their identity was explored through the question: “Have you suffered discrimination because of your religious beliefs, practices or sexual preferences?” Overall, 19% of men and 17% of women reported having experienced this type of discrimination (Table 3). 

### 3.3. Abuse by Mexican Authorities

One of the factors contributing to the gradual impoverishment of those interviewed who were in transit in Mexico was the repeated extortion they suffered by federal and local police, immigration agents, and members of the army located at various checkpoints along the migratory routes. This extortion occurred even when they traveled on commercial buses, colloquially known as “commercial” buses. 

“I think the biggest barrier I have had in Mexico is the police. Yes, they have taken a lot of money from me, the truth is, just to get there, from Hidalgo to Tapachula, how much is that, an hour’s drive? I think they took about twelve thousand pesos from us......to be able to cross into Huiztla, we had to go through three checkpoints and at each checkpoint they took almost 500 pesos from each of us. And that happened even though we were in the COMAR (Mexican Refugee Aid Commission) process. And even though we were told we could go anywhere we wanted in Chiapas” (Omar, Honduran).

Subjects also noted that the authorities frequently ignored, stole, or tore up their humanitarian visas, the permits enabling them to move around the country or authorizing their temporary stay there, issued by Mexican institutions, and which they carried with them. 

“In Querétaro, when we come here…(Tijuana) we also suffered discrimination from the Mexican authorities, who told us that these documents are false (humanitarian visa), and “if we wanted to continue, we had to give them a thousand pesos per head, when we didn’t have any money at all”, Dahlia (Nicaraguan). 

Another sensitive issue was the physical mistreatment and sexual abuse suffered or witnessed by subjects during the searches conducted by the police of commercial bus passengers. 

“…they are with the patrol cars there, and that is when they take the opportunity to get on the bus, take bribes, rob, scam people… a woman sitting next to me had her daughter with her. Both of them were asleep and the officer woke them up. He even checked her private parts, her boobs and all those parts. He began to rummage around, he put his hand down her pants…” Ruth (Honduran).

In the survey administered to migrants, this type of abuse was also explored. A total of 42.2% answered “yes” to the question: “Did you suffer any abuse of authority by immigration agents, police, soldiers, marines, etc.?” A total of 38.4% declared that they had been detained for no reason by an immigration or police authority, while 26.5% stated that the immigration authorities had refused to provide support (see Table 4). 

In addition to this “institutional” violence by the Mexican authorities, there is the risk of becoming victims of organized crime in Mexico.

### 3.4. Violence by Criminal Organizations

In the course of the interviews, we heard reports of this type of violence, and the most recurring sensation or impression in the people who underwent or witnessed them was fear, in addition to other conditions such as the one reported by Salma, the mother of a girl who was assaulted. “We were groped, they groped us, they asked us for everything…they threw us into a ravine, as a result of which my daughter has psychological traumas. Those vandals held a gun to her (pointing to her head)…” Salma (Honduran). 

Subjects who did not directly experience or witness a violent act also said that they felt fear or “nerves” at the possibility of being extorted or attacked by organized crime. Joan, a Honduran, expressed it as, “... mistreatment on the way, a form of kidnapping…”

Carola (Honduran) also remarked that she is frightened when her husband leaves the shelter, “…I’m scared something will happen to him in the street, that they are going to kidnap him because they say that many people are being kidnapped and so on. People are nervous that could happen to them there.” 

The survey data confirm how vulnerable migrants are. As shown in graph 4, over fifty percent experienced this situation during their journey through Mexico. The main types of violence were robbery or assault. They were asked for money or something else and received threats. Men received more threats. Thirteen percent were kidnapped at least once (Table 5).

This contributed to the perception of being permanently in an unsafe context, coupled with the impossibility of leaving or moving elsewhere.

### 3.5. Waiting Time before Being Able to Continue Their Journey

For the subjects, having to spend a long time in a place that was not their final destination made them feel “trapped”, “immobilized”, and “on standby”, which produced “stress”, “sadness”, “helplessness”, “worry” or “anger.” 

“…You feel alone, and shut in, plus the stress of (not knowing) when are you going to leave, when you are going to have your turn, not having any money. Maybe you want to have a soda and if you don’t have friends here, nobody is going to invite you. These are things that make you feel stressed and powerless and vulnerable. You feel tired, you don’t want to go on…” (Diana, Nicaraguan).

As part of the survey, subjects were asked when they had arrived in Tijuana. Of the total number of subjects, 57.3% had spent less than three months there at the time of the interview; 33.2% had arrived six months earlier, and 9.5% had been stranded in this border city for more than six months. 

As shown in Table 6, approximately 60% of the men and women, regardless of the time they had spent in Tijuana, intended to reach the United States. A total of 77.8% of the women who had been in the city for less than three months were in this situation. Furthermore, 22% of all the subjects had been returned after crossing the border, and 12.8% had been returned as part of an asylum seeker program. It is important to note that very few migrants intended to stay in Tijuana.

An association was found between accumulated vulnerability and anxiety symptoms. We confirm that the interaction of various vulnerabilities predisposes individuals to health problems, in this case, emotional discomfort such as anxiety.

In this respect, it was observed that migrants who reported experiencing three or more vulnerabilities presented the highest percentages of anxious symptoms. It is interesting that in the case of men, having two vulnerabilities suffices to triple the prevalence of anxiety, whereas, for women, the presence of three or more vulnerabilities is required to triple this prevalence (Table 7).

## 4. Discussion

Of the five vulnerabilities we identified in our work, four coincide with the risks recognized by González and Aikin [4]. This means that in the past five years, adverse conditions for migrants transiting through Mexico have increased rather than decreased. The new form of vulnerability, according to the subjects, was caused by the “waiting time before being able to continue their journey” or “entrapment”, resulting from the health measures due to the COVID-19 pandemic and the stiffening of containment policies.

After exploring the migration trajectories of the subjects, we considered that the vulnerability that largely determines the way they experience transit through Mexico is “precarious conditions”. These conditions are linked to the reasons they cited for their departure from their countries of origin, which the literature describes as the effects of the persistent inequality and socioeconomic deterioration of the region’s populations [1,2]). This means that individual or family efforts to reverse this type of expulsion by attempting to remain in their places of origin are likely to be unsuccessful. The emotional impact subjects associate with sustained precariousness (such as depression and worry) is experienced from the moment they make the decision to leave their country and continues throughout their journey.

In regard to the vulnerability caused by “rejection and abuse due to their identity”, and according to the contents of the interviews, as well as the survey, we can confirm that discrimination is present in the experiences of migrants. As González and Aikin [4] noted, characteristics such as age, gender, nationality, and skin color conferred greater or lesser vulnerability on people. We were struck by the fact that the impact this kind of rejection had on subjects was expressed by colloquial phrases such as (I felt) “bad”, “I felt awful” that contain a certain degree of imprecision. However, we interpret them in accordance with what Massey [22] and Sennett [23] point out, in the sense that through these phrases, migrants may be expressing the fact that their self-esteem and dignity have been damaged.

From the perspective shared by the subjects and in keeping with what Quesada et al. [8] propose, it is clear how vulnerabilities, “abuse by Mexican authorities”, and “violence by criminal organizations” interact and reinforce each other. As the survey results indicate, over 80% of those surveyed had experienced some form of organized crime violence while in transit in Mexico, and as they recounted in the interviews, this situation mainly caused them to feel fear. Moreover, during their stay in the country, subjects dealt with securitization and containment policies, which translated into border controls, arbitrary detentions, financial extortion and physical abuse. This not only impoverishes them further but, as OHCHR [14] and Cabieses et al. [15] explain, it encourages mistrust in the Mexican authorities, discourages migrants from seeking help and drives them towards riskier routes and modes of transportation, thereby increasing their vulnerability. We believe that this circle of institutional and criminal violence damages emotional and mental health and prevents many migrants who require medical or psychological care from approaching the state institutions that currently provide these services. 

In addition to the above, when the subjects were in “forced immobility” or entrapment [17], some associated this period with feelings of helplessness and episodes of despair that we interpret in keeping the ideas of Requena et al. [21] as an expression of a reduced ability to cope with adversities. We think that for some subjects who experienced the most vulnerabilities, being in a waiting period could increase the risk of presenting with mental disorders or of developing negative coping responses such as the abuse of alcohol or other substances [27].

Studies conducted on the general population have also shown that the likelihood of the onset of mental disorders, including anxiety disorders, rises as the number of adversities or vulnerabilities increases. However, they have also pointed out that there are vulnerabilities that have a greater impact than others and that certain associations are riskier [28,29]. In subsequent studies, it would be useful to explore whether the latter occurs in a similar way in the migrant population. 

As observed in the prevalence of anxiety symptoms and accumulated vulnerability, there is a significant difference by sex. A possible explanation could be found in what was stated by Herrera and Campero [30], who point out that migrant women are continuously exposed to a set of vulnerabilities that they tend to tolerate as a survival strategy. Because of this, emotional discomforts fail to emerge until a greater number of vulnerabilities have accumulated. 

## 5. Conclusions

The use of a mixed methods model made it possible, on the one hand, to describe how a group of migrants accumulate vulnerabilities related to precariousness, identity, institutional abuse, criminal violence, and forced immobility while they are in transit through Mexico and identify which emotional discomforts each of these adversities caused them. At the same time, and based on the statistical data, the model enabled us to state that this type of experience is not exceptional but instead recurrent among Central American migrants. The direct association between mental health and the accumulation of vulnerabilities for a population group could therefore be considered a problem that should be addressed in the public mental health field. 

We confirm the idea expressed by Dahlgren and Whitehead [7] that public mental health must consider the complexity and interdependence of the vulnerabilities experienced by migrants in transit as regards their economic, social, and cultural dimensions.

Although not all migrants who experience these vulnerabilities and anxiety symptoms will develop a severe mental disorder, most of them require humanitarian care to meet their everyday needs, while a significant proportion require psychosocial intervention. 

We believe that some of the strategies proposed by the Lancet Commission on Global Mental Health [29] can be adapted to meet the needs of migrants, such as (1) training people who are in contact with migrant populations in community interventions to help them assess and care for the emotional discomfort triggered by the accumulation of adversity, (2) training first contact personnel (such as migration agents, police officers and military officers) to prevent acts of stigma, discrimination and abuse of power, and (3) expanding actions to provide information on human rights, legal situations, the safety of migratory routes, and other available aid resources. 

The implementation of actions such as these decreases the likelihood of mental disorders occurring or becoming chronic in cases that already had a disorder before migrating. 

Limitations of the study include the fact that results cannot be generalized to all Central American migrants in transit through Mexico since they only correspond to a specific group housed in shelters in Mexico City and Tijuana. Migrants not living in these shelters may share some of the same vulnerabilities but might also face others not explored in this article. Likewise, we consider that although the research investigated a significant number of vulnerabilities, it did not examine all those that may be related to the presence of emotional disorders.

## Figures and Tables

**Table 1 ijerph-20-04899-t001:** Data Collection Methods.

Methods:	Instruments	Selection of Subjects
**Place and date:** Shelters in Mexico City and Tijuana (2021–2022), **Semi-structured individual interviews** Conducted in one session lasting approximately ninety minutes. A space was sought within the shelters where migrants felt comfortable talking. Conducted by one researcher with a Ph.D. in anthropology, one with a master’s in public health and one with a master’s in cultural studies. All the interviews were audio-recorded, transcribed, and stored in digital text files.	**Interview guide:** Topics Migration trajectory (reasons for migrating, obstacles, challenges, and resilience related to migration). Perceptions and beliefs about emotional distress and mental health. Care needs. Service use. Sociodemographic data.	**Intentional sampling ^1^** *Inclusion criteria: (a) over 18 years of age, (b) Central American migrants.* The nationalities of the thirty-five migrants were distributed as follows: twenty-five Hondurans, seven Salvadorans, two Nicaraguans, and one Guatemalan. Eight identified as female, eighteen as male, five as transgender, and three as gay. The ages of the interviewees ranged from seventeen to seventy.
**Place and date:** Shelters in Mexico City and Tijuana (2022). **Survey of migrants** Conducted with migrants who were in the fourteen selected shelters at that time. Interviews were conducted individually (face-to-face). The interviewer asked questions using the Qualtrics software, with the support of tablets.	**Questionnaire** Sections (a) Sociodemographic data: sex, age, place of birth, educational attainment, source of income. (b) Migration trajectory: date they arrived in Mexico and Tijuana, reasons for migrating, related problems, etc. (c) Food insecurity: food with little variety, skipping meals, eating less than they thought they should eat, or feeling hungry, but being unable to eat, and so on). (d) Patient Health Questionnaire-4 (PHQ-4): comprising four questions on a Likert-type scale, with two on the main symptoms of depression and two on anxiety. (e) Scale on violence during displacement through Mexico: robberies, physical assaults, threats, sexual abuse, deprivation of liberty, abuse of authority, etc.	**Sample estimate** To ensure that the results were representative, three visits were made to each of the shelters, and in each one, information was collected on the number of migrants staying at that time, the percentage of women, and the percentage of Central American migrants. Although thirty shelters were located, we worked with a non-random sample of fourteen that agreed to participate. With the information collected, a sample of 250 Central American migrants was obtained with a 95% confidence interval. *Inclusion criteria:* (a) being over 18, (b) Central American migrants. Two hundred and seventeen migrants were surveyed (43.3% men, 53.9% women and 2.8% who declared they were transgender, gender fluid, gay or non-binary). Over 90% were between eighteen and forty-four years old. A total of 43% percent were from Honduras, 22.6% from Guatemala, a similar proportion from El Salvador, and 11.5% from Nicaragua.

^1^ The purpose of intentional sampling is to focus on specific characteristics of the population of interest, in this case, Central American migrants who were living in the shelters and more likely to provide adequate, useful information [25].

**Table 2 ijerph-20-04899-t002:** Food insecurity: during your current stay in Tijuana, did you experience a lack of money or other resources?

	Men		Women		Total		
	(n = 94)		(n = 117)		(n = 211)		
	*f*	*%*	*F*	*%*	*f*	*%*	X^2^/*df*
**They had a limited variety of food**							
No	31	33.0	42	35.9	73	34.6	0.384/1
Yes	63	67.0	75	64.1	138	65.4	
**They ate less than they should**							
No	45	47.9	40	34.2	85	40.3	0.031/1 *
Yes	49	52.1	77	65.8	126	59.7	
**They skipped meals**							
No	39	41.5	48	41.0	87	41.2	0.529/1
Yes	55	58.5	69	59.0	124	58.8	
**They did not eat for a full day**							
No	64	68.1	73	62.4	137	64.9	0.237/1
Yes	30	31.9	44	37.6	74	35.1	

Percentages obtained of total sample by sex. * *p*-value < 0.05.

**Table 3 ijerph-20-04899-t003:** Have you suffered discrimination because of your religious beliefs or sexual practices or preferences?

	Men		Women		Total		
	(n = 94)		(n = 117)		(n = 211)		
	*f*	*%*	*F*	*%*	*F*	*%*	X^2^/*df*
No	76	80.9	97	82.9	173	82.0	0.417/1
Yes	18	19.1	20	17.1	38	18.0	

Percentages obtained of total sample by sex.

**Table 4 ijerph-20-04899-t004:** During your last stay in Mexico, were you?

	Men		Women		Total		
	(n = 94)		(n = 117)		(n = 211)		
	*f*	*%*	*f*	*%*	*f*	*%*	X^2^/*df*
**Refused support by migration authorities**							
No	64	77.8	91	77.8	155	73.5	0.077/1
Yes	30	31.9	26	22.2	56	26.5	
**Arrested for no reason by migratory** **police or military authorities**							
No	49	52.1	81	69.2	130	61.6	0.008/1 *
Yes	45	47.9	36	30.8	81	38.4	
**Subjected to an abuse of authority** **by migratory agents, police, or soldiers**							
No	43	45.7	79	67.5	122	57.8	0.002/1 *
Yes	51	54.3	38	32.5	89	42.2	

Percentages obtained of total sample by sex. * *p*-value < 0.05.

**Table 5 ijerph-20-04899-t005:** During your last stay in Mexico, were you?

	Men		Women		Total		
	(n = 94)		(n = 117)		(n = 211)		
	*f*	*%*	*f*	*%*	*f*	*%*	X^2^/*df*
**Robbed or assaulted**							
No	31	32.9	71	61.7	102	48.3	0.000/1 **
Yes	63	67.1	46	38.3	109	51.7	
**Asked for money or something else**							
No	35	37.2	57	48.7	92	43.6	0.044/1 *
Yes	59	62.8	60	51.3	119	56.4	
**Physically attacked**							
No	63	67.0	94	80.3	157	74.4	0.021/1 *
Yes	31	33.0	23	19.7	54	25.6	
**Threatened**							
No	42	44.6	68	58.1	110	52.1	0.030/1 *
Yes	52	55.4	49	41.9	101	47.9	
**Sexually abused**							
No	93	98.9	111	94.8	204	96.6	0.097/1
Yes	1	1.1	6	5.2	7	3.4	
**Forced to have sex**							
No	90	95.7	110	94.0	200	94.8	0.400/1
Yes	4	4.3	7	6.0	11	5.2	
**Kidnapped**							
No	79	84.0	104	88.9	183	86.7	0.241/1
Yes	15	16	13	11.1	28	13.3	
**A victim of gangs or criminal** **organizations**							
No	73	77.6	97	82.9	170	80.5	0.302/1
Yes	21	22.4	20	17.1	41	19.5	

Percentages obtained of total sample by sex. * *p*-value < 0.05, ** *p*-value < 0.01.

**Table 6 ijerph-20-04899-t006:** Reasons for being in Tijuana and length of stay.

Man	Three Months or Less		Six Months		Over Six Months		Total		
	(n = 67)		(n = 19)		(n = 8)		(n = 94)		
	** *F* **	** *%* **	** *F* **	** *%* **	** *f* **	** *%* **	** *f* **	**%**	** *X^2^/df* **
Returned after crossing	13	19.4	3	15.8	2	25.0	18	19.1	*0.806/3*
Returned as part of an asylum program	12	17.9	3	15.8	0	0	15	16.0	
Planning to go to the United States	37	55.2	11	57.9	6	75.0	54	57.4	
Planning to stay in Tijuana	5	7.5	2	10.5	0	0.0	7	7.4	
**Women**	**Three Months or Less**		**Six Months**		**Over Six Months**		**Total**		
	**(n = 54)**		**(n = 51)**		**(n = 12)**		**(n = 117)**		
	*F*	*%*	*F*	*%*	*f*	*%*	** *f* **	**%**	** *X^2^/df* **
Returned after crossing	7	13.0	15	29.4	4	33.3	26	22.2	*0.011/3 **
Returned as part of an asylum program	2	3.7	10	19.6	3	25.0	15	12.8	
Planning to go to the United States	42	77.8	25	49.0	5	41.7	72	61.5	
Planning to stay in Tijuana	3	5.6	1	2.0	0	0.0	4	3.4	

Percentages obtained of total sample by sex. * *p*-value < 0.05,.

**Table 7 ijerph-20-04899-t007:** Prevalence of anxiety symptoms and accumulated vulnerability.

	Men		Women		Total		
	(n = 33)		(n = 52)		(n = 85)		
	*f*	*%*	*f*	*%*	*f*	*%*	X^2^/*df*
1 vulnerability	4	12.1	7	13.5	11	12.9	0.011/4 *
2 vulnerabilities	13	39.4	6	11.5	19	22.4	
3 vulnerabilities	13	39.4	20	38.5	33	38.8	
4 vulnerabilities or more	3	9.1	18	34.6	21	24.7	

Percentages obtained of the total who presented anxiety symptoms by number of vulnerabilities. * *p*-value < 0.

## Data Availability

Data available on request due to ethical restrictions.

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
