# Peer review of "Anxiety among Central American Migrants in Mexico: A Cumulative Vulnerability"

_ijerph, 2023, doi:10.3390/ijerph20064899_

Round 1

Reviewer 1 Report

The topic of the article is relevant. The use of mixed-methods study (QUALI-QUAN) is a huge advantage of this study. The authors obtained interesting results and described them well. But, unfortunately, the authors did not specifically show how the data they obtained can be used for psychological assistance to migrants. Perhaps the results of this study can be used to improve the activities of social and psychological services that work with migrants. The authors should have provided more specific information on the practical significance of the results.

Author Response

We are grateful for the comments on the manuscript “Stress and Anxiety among Central American Migrants in Mexico: a cumulative vulnerability.” We have responded to the comments and suggestions of the reviewers in the document they sent us. We have also included the responses to each one in the section requested.

Revisor 1:

The topic of the article is relevant. The use of mixed-methods study (QUALI-QUAN) is a huge advantage of this study. The authors obtained interesting results and described them well. But, unfortunately, the authors did not specifically show how the data they obtained can be used for psychological assistance to migrants. Perhaps the results of this study can be used to improve the activities of social and psychological services that work with migrants. The authors should have provided more specific information on the practical significance of the results.

R= In the discussion section, we have included a paragraph on the importance of the results for improving social and psychological services for the care of migrant populations in transit through Mexico.

Reviewer 2 Report

The article Stress and Anxiety among Central American Migrants in Mexico: a cumulative vulnerability, deals with a pertinent issue regarding the mental health of immigrants. 

The article is well grounded and the methods are appropriate. However the presentation of the results is descriptive, and the quantitative data could be better explored if illustrated in table instead of graphical formats. It would also be relevant to perform statistical analyses that make correlations between the variables. The  data could be presented in tables and not in graphs. 

The data reveals that migrants suffer from multiple problems. The authors argue that there is an association between cumulative vulnerability and anxiety symptoms, but there is no evidence in the data presented that this is the case. A statistical analysis would need to be carried out to correlate or demonstrate unequivocally that this is the case. 

Another issue that deserves revision is the title of the article, since it talks about stress, and anxiety symptoms are related whit stress, but the data collected are lacking in relation to this argument. The bibliography is adequate.

Author Response

We are grateful for the comments on the manuscript “Stress and Anxiety among Central American Migrants in Mexico: a cumulative vulnerability.” We have responded to the comments and suggestions of the reviewers in the document they sent us. We have also included the responses to each one in the section requested.

Revisor 2:

However, the presentation of the results is descriptive, and the quantitative data could be better explored if illustrated in table instead of graphical formats.

R= We have changed the format from graphs to tables and complemented the information with the statistical analyses carried out.

It would also be relevant to perform statistical analyses that make correlations between the variables. The  data could be presented in tables and not in graphs.

R= The statistical analyses performed were included in the tables  

The data reveals that migrants suffer from multiple problems. The authors argue that there is an association between cumulative vulnerability and anxiety symptoms, but there is no evidence in the data presented that this is the case. A statistical analysis would need to be carried out to correlate or demonstrate unequivocally that this is the case.

R= The table version included the statistical analysis carried out and the significance level obtained (0.011). This corroborates the association between the presence of anxiety symptoms and the accumulation of vulnerabilities.

Another issue that deserves review is the title of the article, since it talks about stress, and anxiety symptoms are related whit stress, but the data collected are lacking in relation to this argument

R= In keeping with this comment, we suggest the following title for the article: Anxiety among Central American Migrants in Mexico: a cumulative vulnerability